# DUAL SKEW DIVERGENCE LOSS FOR NEURAL MACHINE TRANSLATION

## ABSTRACT

For neural sequence model training, maximum likelihood (ML) has been commonly adopted to optimize model parameters with respect to the corresponding objective. However, in the case of sequence prediction tasks like neural machine translation (NMT), training with the ML-based cross entropy loss would often lead to models that overgeneralize and plunge into local optima. In this paper, we propose an extended loss function called *dual skew divergence* (DSD), which aims to give a better tradeoff between generalization ability and error avoidance during NMT training. Our empirical study indicates that switching to DSD loss after the convergence of ML training helps the model skip the local optimum and stimulates a stable performance improvement. The evaluations on WMT 2014 English-German and English-French translation tasks demonstrate that the proposed loss indeed helps bring about better translation performance than several baselines.

## 1   INTRODUCTION

Neural machine translation (NMT) (Kalchbrenner & Blunsom, 2013; Sutskever et al., 2014) has shown remarkable performance in terms of sequence-to-sequence learning framework for diverse language pairs. Unlike the conventional statistical machine translation (SMT) (Koehn et al., 2003) that explicitly models linguistic features of the training corpus, NMT aims at building an end-to-end model that directly transforms a source language sequence to the target one. With the development of attention mechanism (Bahdanau et al., 2015; Luong et al., 2015), NMT has achieved results on par with or even better than SMT.

During NMT training, maximum likelihood (ML) is the most commonly used strategy that maximizes the likelihood of the target sentence conditioned on the source throughout training corpus. In practice, ML-based loss is often represented with a word-level cross entropy form, which has proven to be effective for NMT modeling. However, Ranzato et al. (2016) pointed out that ML training suffers from two drawbacks. First, the model is only exposed to training distribution and ignores its own prediction errors during training. Second, a word-level loss is used to optimize the model parameters at the training time while at inference the model prediction is evaluated by a sequence-level metric such as BLEU (Papineni et al., 2002). To handle such problems, several recent work focused on the research of more effective and direct training strategies. Bengio et al. (2015) advocated a *curriculum learning* approach that gradually forces the model to deal with its own mistakes as it does during inference. Wiseman & Rush (2016) proposed a sequence-level loss function in terms of errors made during beam search. Shen et al. (2016) applied minimum risk training (MRT) from SMT to optimize NMT modeling directly in terms of BLEU score. Some other work resorted to reinforcement learning based approaches (Ranzato et al., 2016; Bahdanau et al., 2017).

In this work, we introduce a novel loss called *dual skew divergence* (DSD) to compensate for the original ML-based loss. It can be proved that maximizing likelihood is equal to minimizing the Kullback-Leibler (KL) divergence (Kullback & Leibler, 1951) $KL(P||Q)$ between the real data distribution $P$ and the model prediction $Q$. According to Huszár (2015), minimizing $KL(P||Q)$ tends to find a $Q$ that covers all the true data distribution and ignores the rest of incorrect candidates, which will lead to models that overgeneralize and generate implausible samples during inference. Meanwhile, minimizing the opposite form $KL(Q||P)$ has a distinct property that tends to model a $Q$ that avoids assigning probability mass to the unlikely terms. To benefit from both of these two

divergence and balance the tradeoff between the original ML principle and the concern about the undesirable model prediction, we interpolate $KL(P||Q)$ and $KL(Q||P)$ to form a new DSD loss.

We carry out all the experiments on both English-German and English-French translation tasks from WMT 2014 and compare our models to some other work with similar model size and dataset. The early experiments indicate that switching to DSD loss after the convergence of ML training, a simulated annealing mechanism is actually introduced and it indeed helps the model jump out a local optimum within a short time. Evaluations on the test sets show that our DSD-extended models are better than the ML-only ones and outperform a series of strong baselines significantly.

## 2 NEURAL MACHINE TRANSLATION

In this paper, we closely follow the neural machine translation model proposed by Bahdanau et al. (2015).[1] Different from the conventional statistical models, it is mainly based on a neural encoder-decoder network with attention mechanism (Bahdanau et al., 2015; Luong et al., 2015). In this section, we will give a brief introduction about this basic architecture.

The encoder is a bidirectional recurrent neural network (RNN) such as Gated Recurrent Unit (GRU) (Cho et al., 2014) or Long Short-Term Memory (LSTM) (Hochreiter & Schmidhuber, 1997). The forward RNN reads an input sequence $x = (x_1, ..., x_m)$ from left to right and calculates a forward sequence of hidden states $(\overrightarrow{h}_1, ..., \overrightarrow{h}_m)$ as the representation of the source sentence. Similarly, the backward RNN reads the input sequence in a reverse direction and learns a backward sequence $(\overleftarrow{h}_1, ..., \overleftarrow{h}_m)$. The hidden states of the two RNNs $\overrightarrow{h}_i$ and $\overleftarrow{h}_i$ are concatenated to obtain the source annotation vector $h_i = [\overrightarrow{h}_i, \overleftarrow{h}_i]^T$ as the initial state of the decoder.

The decoder is a forward RNN that predicts a corresponding translation $y = (y_1, ..., y_n)$ step by step. The translation probability can be formulated as follows:

$$p(y_j|y_{<j}, x) = q(y_{j-1}, s_j, c_j),$$

where $s_j$ and $c_j$ denote the decoding state and the source context at the $j$-th time step respectively. Here, $q(\cdot)$ is the softmax layer and $y_{<j} = (y_1, ..., y_{j-1})$. Specifically,

$$s_j = g(y_{j-1}, s_{j-1}, c_j),$$

where $g(\cdot)$ is the corresponding RNN unit. The context vector $c_j$ is calculated as a weighted sum of the source annotations $h_i$ on the basis of the attention mechanism:

$$c_j = \sum_{i=1}^{m} \alpha_{ji} h_i.$$

The alignment model $\alpha_{ji}$ defines the probability that how well $y_j$ is aligned to $x_i$, which is a single layer feed-forward neural network.

The whole model is jointly trained to seek the optimal parameters that can be used to correctly encode the source sentences and decode them to the corresponding target sentences.

## 3 DUAL SKEW DIVERGENCE LOSS FOR NMT

### 3.1 CROSS ENTROPY

In information theory, cross entropy is an important concept measuring the difference between two probability distributions. In natural language processing, cross entropy is usually used as the evaluation metric of language modeling. During NMT training, word-level cross entropy broadly serves as the loss function to learn the model parameters.

The NMT model aims at training a single, large neural network that directly transforms a given source sequence $x = (x_1, x_2, ..., x_m)$ to the corresponding target sequence $y = (y_1, y_2, ..., y_n)$.

---

[1]During this study, we are aware that there come advanced models such as Transformer (Vaswani et al., 2017) and ConvS2S (Gehring et al., 2017). To focus on the loss improvement, we still keep a simple architecture as baseline in this paper.

Given a set of training examples $D = \{\langle x^{(s)}, y^{(s)} \rangle\}_{s=1}^{S}$, the standard ML training objective is to maximize the log-likelihood of the training data with respect to the parameters $\boldsymbol{\theta}$:

$$\hat{\boldsymbol{\theta}}_{\text{ML}} = \arg\max_{\boldsymbol{\theta}}\{\mathcal{L}(\boldsymbol{\theta})\},$$

$$\mathcal{L}(\boldsymbol{\theta}) = \sum_{s=1}^{S} \log P(y^{(s)}|x; \boldsymbol{\theta}) = \sum_{s=1}^{S} \sum_{i=1}^{n} \log P(y_i^{(s)}|x^{(s)}, y_{<i}^{(s)}; \boldsymbol{\theta}).$$

To fit the gradient descent method, the objective is actually transformed to minimize the negative log-likelihood. In practice, it is realized by minimizing the word-level cross entropy that can be calculated at each time step of decoding. Given an observed target sequence of length $n$, the cross entropy loss in the vector form can be represented as

$$J_{\text{XENT}} = -\sum_{i=1}^{n} \mathbf{y_i} \log(\hat{\mathbf{y}_i}), \tag{1}$$

where $\mathbf{y_i}$ is a one-hot vector, referring to the correct target label at time step $i$ and $\hat{\mathbf{y}_i}$ is the model approximation given by the softmax layer.

## 3.2 KULLBACK-LEIBLER DIVERGENCE

Kullback-Leibler (KL) divergence (Kullback & Leibler, 1951) is also a measurement that calculates the distance between two random probability distributions denoted by:

$$D_{KL}(P||Q) = E_{x \sim P}[\log P(x) - \log Q(x)], \tag{2}$$

where $P$ is the true data distribution of target words and $Q$ is the distribution of model prediction.

Mathematically, there is some connection between KL divergence $D_{KL}(P||Q)$ and cross entropy $H(P, Q)$, which can be described as follows:

$$H(P, Q) = H(P) + D_{KL}(P||Q),$$

where $H(P)$ refers to the entropy of data itself. Since the gradient of $H(P)$ with respect to model parameters about $Q$ is always equal to zero, training with $H(P, Q)$ is identical to that with $D_{KL}(P||Q)$. In other words, minimizing cross entropy loss actually means minimizing KL divergence.

## 3.3 ASYMMETRY OF KL DIVERGENCE

Exchanging the position of $P$ and $Q$, we can easily get $D_{KL}(Q||P)$ that is in an inverse direction with $D_{KL}(P||Q)$. Following Eq.(2), $D_{KL}(Q||P)$ can be also rewritten as

$$D_{KL}(Q||P) = E_{x \sim Q}[-\log P(x) + \log Q(x)].$$

Obviously, $D_{KL}(P||Q)$ is not identical to its inverse form $D_{KL}(Q||P)$. Huszár (2015) showed that minimizing $D_{KL}(P||Q)$ tends to find a $Q$ that covers all modes (peak value) of $P$ at the cost of placing probability mass where $P$ has none. However, minimizing $D_{KL}(Q||P)$ leads to a mode-seeking behavior: the optimal Q will concentrate around the largest mode of $P$ while completely ignore smaller modes.

In the case of NMT, multiple target candidate words could have the same high probabilities at each step of decoding. Consequently, we may have more than one best choice at the same time. Thereby, the different effects of training with a word-level $D_{KL}(P||Q)$ or $D_{KL}(Q||P)$ lie in two aspects: First, the model trained by minimizing $D_{KL}(P||Q)$ will tend to memorize all possible outputs and some words with low probabilities in $P$ will also be assigned high probabilities; Second, the model trained by minimizing $D_{KL}(Q||P)$ will instead try to memorize only a subset of all possible outputs, while the cost is, some words with high probabilities in $P$ will be ignored. In general, the two different objectives have their own advantages and disadvantages, so it is hard to say which is actually better. Despite of this, it still provides us with a new direction of choosing the proper loss function.

### 3.4 Skew Divergence

The analysis above shows that $D_{KL}(Q||P)$ may be also a good alternative loss function for NMT training. However, $D_{KL}(Q||P)$ is only well-defined when $P$ is never equal to zero, which is not guaranteed in the NMT training. To overcome the inconvenience, we use some kind of approximations instead. One such approximation function is the $\alpha$-*skew divergence* (Lee, 1999):

$$s_\alpha(Q, P) = D_{KL}(Q||\alpha Q + (1 - \alpha)P),$$

where $\alpha$ controls the degree to which the function approximates $D_{KL}(Q||P)$ and $0 \leq \alpha \leq 1$.

By assigning a small value to $\alpha$, we can simulate the behavior of minimizing $D_{KL}(Q||P)$ with $\alpha$-*skew divergence* approximation. In our experiments, $\alpha$ is set to 0.01 constantly.

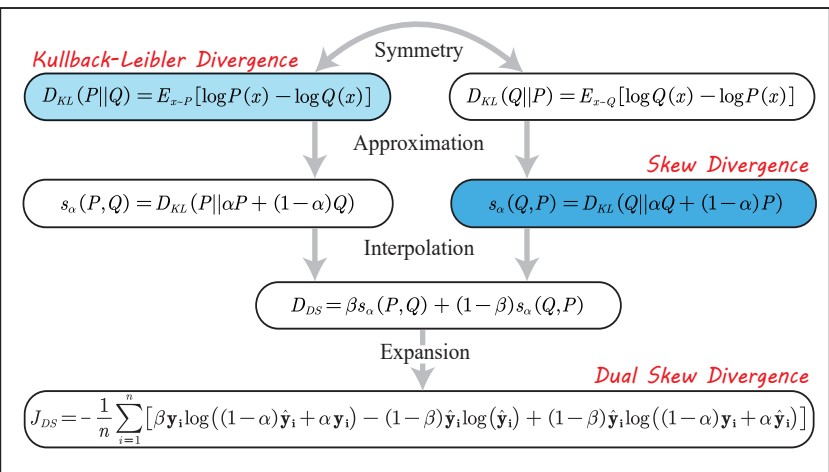

Figure 1: The general idea of the proposed DSD loss function.

### 3.5 Dual Skew Divergence

Finally, we introduce a new loss with the expectation of retaining the good part of ML objective and meanwhile compensating its missing part. In order to obtain a symmetrical form of the loss, we also approximate $D_{KL}(P||Q)$ with $s_\alpha(P, Q)$. By interpolating between both directions of $\alpha$-*skew divergence*, we derive the new loss function, which will be referred to as *dual skew divergence* (DSD) below. The interpolated DSD function is given by

$$D_{DS} = \beta s_\alpha(P, Q) + (1 - \beta)s_\alpha(Q, P), \tag{3}$$

in which $\beta$ is the interpolation coefficient and $0 \leq \beta \leq 1$.

In terms of notations in Eq.(1), we rewrite Eq.(3) in a computationally implementable form as below:

$$J_{DS} = -\frac{1}{n}\sum_{i=1}^{n}[\beta\mathbf{y_i}\log((1 - \alpha)\hat{\mathbf{y_i}} + \alpha\mathbf{y_i}) - (1 - \beta)\hat{\mathbf{y_i}}\log(\hat{\mathbf{y_i}}) + (1 - \beta)\hat{\mathbf{y_i}}\log((1 - \alpha)\mathbf{y_i} + \alpha\hat{\mathbf{y_i}})],$$

where $n$ is the length of target sequence and the term about $H(P)$ is conveniently omitted since it will not appear in the gradient as discussed above. Note that while it is well-defined in mathematics, we still need to add a small enough constant ($10^{-12}$ in our implementation) to each log term for numerical stability. Our main idea about the derivation of the proposed DSD loss is summarized in the diagram of Figure 1 step by step.

## 4 Experiments

### 4.1 Data Preparation

We perform all the experiments on data from the shared task of WMT 2014 and report the results on both English-German and English-French translation tasks. The translation quality is measured by

case-sensitive 4-gram BLEU score (Papineni et al., 2002) and we use sign test (Collins et al., 2005) to test the statistical significance of our results.

For English-German task, the training set consists of 4.5M sentence pairs with 91M English words and 87M German words. For English-French task, the training set contains 12M sentence pairs with 304M English words and 348M French words. The models are evaluated on the WMT 2014 test set *news-test* 2014, and the concatenation of *news-test* 2012 and *news-test* 2013 is used as the development set.

The preprocessing on both training sets includes a joint byte pair encoding (Sennrich et al., 2016) with 32K merge operations after tokenization. The final joint vocabulary size is around 37K for English-German and 37.2K for English-French translation task respectively. Every out-of-vocabulary word is replaced with a special ⟨UNK⟩ token.

## 4.2 MODELS AND TRAINING DETAILS

We train two different baselines with SMT and NMT systems respectively. For the SMT baseline, we use the phrase-based SMT system MOSES (Koehn et al., 2007). The log-linear model of MOSES is trained by the minimum error rate training (MERT) (Och, 2003) that directly optimizes model parameters with respect to evaluation metrics. Our SMT baseline is trained with the default configurations in MOSES together with a trigram language model trained on the target language using SRILM (Stolcke, 2002). For the NMT baseline, we use the model architecture of the attention-based RNNSearch (Bahdanau et al., 2015). Our NMT baseline model is generally similar to Bahdanau et al. (2015), except that the input feeding approach is applied and the attention layer is built on top of LSTM layer instead of GRU.

During NMT training, each direction of the LSTM encoder and the LSTM decoder are with 1000 dimensions. The word embedding and the attention size is set to 620. The batch size is set to 128, and no dropout is used for all models. The training set is reshuffled at the beginning of each epoch. A single Nvidia Titan X (Pascal) GPU is used to train all the NMT models.

For English-German task, training lasts for 9 epochs in total. We use Adam optimizer for the first 5 epochs with the learning rate of $3.0 \times 10^{-4}$, and then switch to plain SGD with the learning rate of 0.1. At the beginning of epoch 8, we decay the learning rate to 0.05.

For English-French task, the models are trained for 4 epochs. Adam optimizer with the learning rate of $3.0 \times 10^{-4}$ is used for the first 2 epochs. We then switch to SGD with the learning rate of 0.1, and finally decay the learning rate to 0.05 at the beginning of epoch 4.

To demonstrate the effectiveness of our model, we also compare it to several state-of-art NMT systems with the same dataset and similar model size.

- RNNSearch-LV (Jean et al., 2015): a modified version of RNNSearch based on importance sampling which allows the model to have a very large target vocabulary without any substantial increase in computational cost.
- Local-Attn (Luong et al., 2015): applying a local attention mechanism that focuses only on a small subset of the source positions when predicting each target word.
- MRT (Shen et al., 2016): optimized by a loss function for minimum risk training. The model parameters are directly optimized with respect to the evaluation metrics.
- Bahdanau-LL (Bahdanau et al., 2017): this model closely followed the architecture of Bahdanau et al. (2015) with ML training and achieved a higher performance by annealing the learning rate and penalizing too short output sequences during beam search.
- Bahdanau-AC+LL (Bahdanau et al., 2017): a neural sequence prediction model that combines the actor-critic from reinforcement learning with the original ML training.

For DSD training, we keep the hyper-parameter settings the same as NMT baselines. However, we tried a series of preliminary experiments and found that DSD training alone from random initialization would be easy to cause vanishing gradient failure with stuck on low BLEU scores. Thus we eventually adopt a hybrid training strategy. To provide a reliable initialization, the training starts from cross entropy loss and then switch to DSD loss at different switching points. The model trained with such strategy is referred to DSD-NMT hereafter.

### 4.3 EFFECT OF $\beta$

As shown in Section 3.5, the interpolation parameter $\beta$ controls the degree of conservativeness of the model. When $\beta$ is close to one, the loss function behaves more like $D_{KL}(P||Q)$ which tends to cover all the modes of $P$. When $\beta$ is close to zero, the loss function behaves more like $D_{KL}(Q||P)$ which prefers to memorize only a subset of the true data distribution conservatively. In order to find an optimal value of $\beta$, we study the effect of $\beta$ on the translation quality of English-German task. Table 1 reports the BLEU scores with different $\beta$ on the development set with greedy search. It shows that the model trained with $\beta = 0$ performs the best, which surprisingly indicates that the skew inverse KL divergence is better than the interpolation form in terms of our loss function applying strategy. Note that the inverse form of skew divergence loss is still a special case of our general DSD loss.

Table 1: BLEU scores with different $\beta$ on English-German dev set

| Model | Baseline | $\beta = 1$ | $\beta = 0.5$ | $\beta = 0$ |
|---|---|---|---|---|
| BLEU | 20.11 | 20.41 | 20.94 | **21.32** |

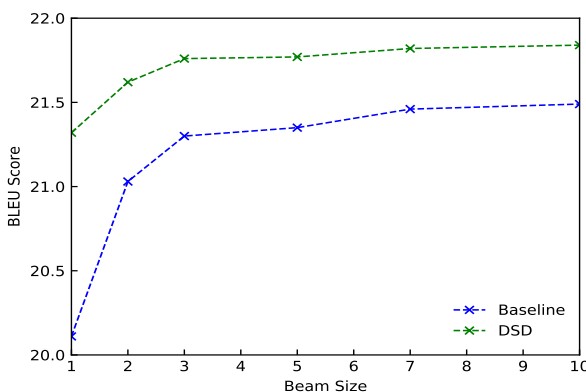

Figure 2: BLEU scores with different beam sizes on English-German dev set($\beta = 0$).

### 4.4 EFFECT OF BEAM SIZE

In Figure 2, we plot the variation curves of BLEU scores for the proposed DSD model and NMT baseline with different beam sizes on the English-German development set. With the increasing of beam size, the BLEU score is also increasing and the best score is given by the beam size of 10 for both NMT baseline and DSD-NMT. However, when beam search is used, the margin between the proposed DSD method and ML training becomes smaller, which is a general observation for beam search is more tolerable to temporary mistakes.

### 4.5 TRAINING LOSS AND BLEU

Since we start the training of DSD-NMT with an initial ML learning, it is meaningful to study how much the model performance with this training strategy will be influenced by the switching location timing. In Figure 3, we plot the curves of BLEU scores and training loss against training steps from the actual training process for English-German task with greedy search at different switching positions (steps of 100K, 200K, 245K and 313K). Figure 3 shows that when switching to our new loss function, the BLEU scores are all improved, especially, there is more than one point improvement at steps 245K and 313K. Comparing all the training curves shows that a better DSD switching point should be located around the convergence of the standard ML training. At the same time after DSD switching, the cross entropy loss that is supposed to be minimized keeps increasing as the corresponding BLEU scores, which undoubtedly indicates that cross entropy fails to reflect translation quality near the end of its training. The long time training may cause the model to plunge into a lo-

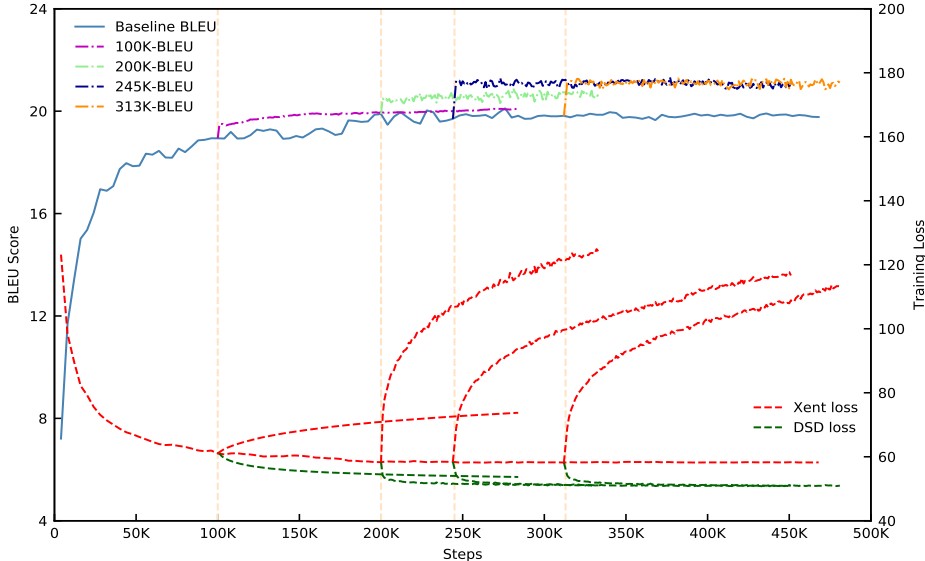

Figure 3: BLEU scores and training loss on English-German dev set with DSD switching after 100K, 200K, 245K and 313K steps ($\beta = 0$, greedy search).

cal optimum, while the loss switching operation brings a model behavior like a simulated annealing mechanism[2] and helps the model jump onto a better optimum.

Table 2: Performance on English-German task

| Model | Method | BLEU |
|---|---|---|
| RNNsearch-LV | ML+beam | 19.40 |
| Local-Attn | ML+beam | 20.90 |
| MRT | MRT+beam | 20.45 |
| Baseline-SMT | MERT+greedy | 18.83 |
| | MERT+beam | 19.91 |
| Baseline-NMT | ML+greedy | 20.89 |
| | ML+beam | 22.13 |
| | ML+deep | 24.64 |
| DSD-NMT | DSD+greedy | **22.02**[++] |
| | DSD+beam | **22.60**[+] |
| | DSD+deep | **25.00**[++] |

Table 3: Performance on English-French task

| Model | Method | BLEU |
|---|---|---|
| RNNsearch-LV | ML+beam | 34.60 |
| MRT | MRT+beam | 34.23 |
| Bahdanau-LL | ML+greedy | 29.33 |
| | ML+beam | 30.71 |
| Bahdanau-AC+LL | ML+AC+greedy | 30.85 |
| | ML+AC+beam | 31.13 |
| Baseline-SMT | MERT+greedy | 31.55 |
| | MERT+beam | 33.82 |
| Baseline-NMT | ML+greedy | 32.10 |
| | ML+beam | 34.70 |
| DSD-NMT | DSD+greedy | **33.56**[++] |
| | DSD+beam | **35.04**[+] |

## 4.6 RESULTS ON TEST SETS

The results about English-German and English-French translation tasks on test sets are reported in Tables 2 and 3.[3] For previous work, the best BLEU scores of single models are listed from the original papers. From Tables 2 and 3, we see that DSD-NMT outperforms all the other models and our own baselines using standard cross entropy loss with both greedy search and beam search.

For English-German task, Table 2 shows that even our baseline NMT models achieve better performance than most of the listed systems. It may owe to the integrative action of joint BPE, input feeding and the mixed training strategy of Adam and SGD algorithms. For greedy search, our DSD model outperforms SMT and NMT baselines by 3.19 and 1.13 BLEU respectively. It is even better

---

[2]We also tried an automatically loss function switching strategy just like simulated annealing that switches the loss according to the growth rate of BLEU. However, switching DSD back to cross entropy will not furthermore increase BLEU over the original score.

[3]"++" indicates statistically significant difference from NMT baseline at $p < 0.01$ and "+" at $p < 0.05$.

than the best listed system Local-Attn (Luong et al., 2015) with beam search. For beam search, DSD-NMT outperforms SMT baseline with improvement of 2.69 BLEU points. However, it increases only 0.47 BLEU over NMT baseline, which is in line with our discussion in Section 4.4. Furthermore, we also test our DSD loss on a deep NMT model where the encoder and decoder are both stacked 4-layer LSTMs. The result also shows that we can get 0.36 points improvement which demonstrates the effectiveness and robustness of our DSD model.

For English-French task, with greedy search, the performance of DSD-NMT is still superior to other systems listed in Table 3. It achieves an increase of 1.46 and 2.01 BLEU points for NMT and SMT baselines respectively. For beam search, our DSD-NMT outperforms SMT and NMT baselines by 1.22 and 0.34 BLEU, respectively. Compared to other previous models, DSD-NMT with greedy search also brings about more performance improvement than beam search.

The results of the sign test in both tables also indicate that DSD-NMT indeed enhances the translation quality over baselines trained only with cross entropy loss at high significant levels.

## 5 RELATED WORK

The standard NMT system commonly adopts the word-level cross entropy loss to learn model parameters. However, this type of ML learning has been shown not to be an optimal method for sequence model training (Ranzato et al., 2016). A number of recent work attempted to seek different training strategies or improve the loss function. One of such approaches is by Bengio et al. (2015), who proposed to gently change the training process from a fully guided scheme using the true previous token towards a less guided scheme which mostly uses the generated tokens instead. Some other researches focused on the study of sequence-level training algorithm. For instance, Shen et al. (2016) applied minimum risk training (MRT) in end-to-end NMT. Their basic idea is to minimize the expected loss with respect to the posterior distribution and the model parameters are directly optimized according to the evaluation metrics. Wiseman & Rush (2016) introduced a sequence-level loss function in terms of errors made during the beam search. They develop a non-probabilistic variant of seq2seq model that can assign a score to any possible target sequence. It avoids classical biases associated with local training and unifies the training loss with the test-time usage. Another sequence-level training algorithm was proposed by Ranzato et al. (2016) that directly optimizes the metrics used at test time. It is built on REINFORCE algorithm. Their major contribution is Mixed Incremental Cross-Entropy Reinforce (MIXER) based on incremental learning and the use of a hybrid loss function which combines both REINFORCE and cross entropy. Similarly, with a reinforcement learning style scheme, Bahdanau et al. (2017) introduced a *critic* network to predict the value of an output token, given the policy of an *actor* network. This results in a training procedure that is much closer to the test phase, and allows the model to directly optimize for a task-specific score such as BLEU. Edunov et al. (2018) presented a comprehensive comparison of classical structured prediction loss for seq2seq models. Differently, we intend to optimize the training loss considering both easy implementation and not introducing more model complexities.

## 6 CONCLUSION

This work proposes a general and balanced loss function called *dual skew divergence* for NMT training. Adopting a hybrid training strategy with both cross entropy and DSD training, we empirically verify that switching to DSD loss after the convergence of ML training can introduce a simulated annealing behavior to stably help the model jump onto a better optimum. More exactly, our studies show that skew inverse KL divergence (deep blue formula in Figure 1) working together with KL divergence (i.e., cross entropy loss, shallow blue formula in Figure 1) actually gives the best performance in terms of the general using of DSD loss. The evaluations on WMT 2014 English-German and English-French translation tasks prove that the proposed novel loss indeed improves ML-only training and outperforms other strong baselines.

Since the DSD loss can be generalized to apply in all other seq2seq models, it is promising to extend this proposed approach to other related tasks, such as text summarization and machine reading comprehension. We are also trying our best to apply DSD loss to deeper and stronger models like Transformer (Vaswani et al., 2017) and ConvS2S (Gehring et al., 2017) which are more sensitive to model parameters.

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
