# OpenReview forum: "Dual Skew Divergence Loss for Neural Machine Translation"
_ICLR.cc/2019/Conference_

### Official Review · AnonReviewer1 · 2018-10-31
**Interesting idea, results are not quite there yet**

**Rating:** 5
**Confidence:** 4

**Review:**

This paper describes an alternative training objective to cross-entropy loss for sequence-to-sequence models. The key observation is that cross-entropy is minimizing KL(P|Q) for a data distribution P and a model distribution Q; they add another loss that minimizes the inverse KL(Q|P) to create their dual-skew divergence. The idea is tested in the context of neural MT, using a model similar to that proposed by Bahdanau et al. (2015) with results on English-to-French and English-to-German WNT 2014. In the context of beam search, improvements are small (<=0.5 BLEU) but statistically significant.

This is an interesting idea, and one I certainly wouldn’t have thought of on my own, but I think it is currently lacking sufficient experimental support to warrant publication. The paper feels strangely dated, with most experiments on two-layer models, and only two citations from 2017. The experiments compare against an in-house maximum likelihood baseline (varying greedy-vs-beam search and model depth), and against a number of alternative training methods (minimum risk, scheduled sampling, RL) with numbers lifted from various papers. These latter results are not useful, as the authors (helpfully) point out that the baseline results in this paper are universally higher than the baselines from these other papers. Furthermore, it feels like methods designed to address exposure bias and/or BLEU-perplexity mismatch are not the right comparison points for this work, as it does not attempt to address either of these issues. I would instead be much more interested to see a comparison to label smoothing (Szegedy et al., 2015), which perhaps addresses some of the same issues, and which produces roughly the same magnitude of improvements. Also, the literature review should likely be updated to include Edunov et al., 2017. In general, the improvements are small (though technically statistically significant), the baseline models are somewhat shallow and the deltas seem to be decreasing as model depth grows, so it is hard to get too excited.

Smaller concerns:

For Table 1, it would be helpful to explain why Baseline is not equal to \Beta=1. With some effort, I figured out that this was due to the alpha term modifying the cross-entropy objective when \Beta=1.

It would also be useful to tell us what “switching point” was used for Table 1 and Figure 2.

Christian Szegedy, Vincent Vanhoucke, SergeyIoffe, Jonathon Shlens, and Zbigniew Wojna. 2015. Rethinking the inception architecture for computer vision. CoRR abs/1512.00567. http://arxiv.org/abs/1512.00567.

Sergey Edunov, Myle Ott, Michael Auli, David Grangier, and Marc’Aurelio Ranzato. 2018. Classical structured prediction losses for sequence to sequence learning. In Proceedings of NAACL-HLT 2018.

---

### Official Review · AnonReviewer2 · 2018-11-02
**Relatively low score on novelty but high score on experimental implementation**

**Rating:** 6
**Confidence:** 4

**Review:**

This paper presents a new loss objective for NMT. The main idea is to optimize an interpolation of KL(P|Q) and KL(Q|P), which is Kulback-Liebler Divergence computed at the word-level for model distribution Q and true distribution P. The motivation is that KL(P|Q) finds a Q that covers all modes of the data whereas KL(Q|P) finds a Q that concentrates on a single mode. So optimizing on the interpolation gets the best of both worlds. In my opinion, this is a relatively simple and known idea in ML (but perhaps not in MT? I'm not sure.) On the other hand, the NMT experiments are well-implemented and convincingly shows that it improves BLEU on a WMT dataset.

In general, the experiments look solid. I applaud the multiple baseline implementations, in particular even including the SMT baseline. The lack of transformer/CNN models is not a demerit in my opinion, since the focus is on loss objectives and the LSTM models are just as reasonable.

The paper is clearly written, with a few exceptions. It is not clear why you have to first train with ML before switching to the proposed DSD objective. As such, Section 4.5 should be prefaced with a motivation. Also, Figure 3 is hard to read with the two kinds of plots -- maybe split into two figures?

An open question is: does your model capture the issues of mode covering as mentioned in the motivation? It would be helpful to include analyses of the word-level distributions to quantify the differences (e.g. word entropy) between ML and various KL/DSD solutions. Also I would recommend showing train/test set perplexity scores of the various proposed and baseline methods.

As a minor point for argumentation: it is not clear that your proposal addresses the sequence-level loss vs word-level loss issue. It is conceivable, but it seems indirect and there is no quantifiable connection between the word-level loss (such as DSD) and a sequence-level loss. Or is there?

---

### Official Review · AnonReviewer3 · 2018-11-05
**An idea worth exploring but the paper has flaws.**

**Rating:** 3
**Confidence:** 4

**Review:**

The paper describes a new loss function for training, that can be
used as an alternative to maximum likelihood (cross entropy), or
as a metric that is used to fine-tune a model that is initially
trained using ML.

Experiments are reported on the WMT 2014 English-German and
English-French test sets.

I think this is an idea worth exploring but overall I would not
recommend acceptance. I have the following reservations:

* I found much of the motivation/justification for the approach
unconvincing - too heuristic and informal. What does it mean
to "overgeneralize" or "plunge into local optima"? Can we say
anything semi-formal about this alternative objective?

* The improvements over ML are marginal, and there are a lot of moving
parts/experimental settings in these models, i.e., a lot of
tweaking. The results in tables 2 and 3 show a 0.36/0.34 improvement
over ML using DSD. (btw, what is meant by "DSD-deep" or "ML-deep"? I'm
not sure these terms are explained?)

* The comparison to related work is really lacking. The "Attention is
all you need" paper (Vaswani et al.) reports 28.4/41.0 BLEU for these
test sets, respectively 3.4/5.96 BLEU points better than the results
in this paper. That's a huge gap. It's not clear that the improvements
(again, less than 0.5 BLEU points) will remain with a state-of-the-art
system. And I think the paper is misleading in how it cites previous
results on these data sets - there is no indication in the paper that
these better results are in the literature.

Some small things:

* unplausible -> implausible

* "Husz (2015) showed that D(P || Q) is not identical to its inverse form
D(Q || P)" this is well known, predating 2015 for sure.

---

### Meta-Review · Area_Chair1 · 2018-12-13
**Interesting, but heuristic idea. Experiments somewhat unconvincing.**

**Confidence:** 4
**Recommendation:** Reject

**Metareview:**

This paper proposes a new loss function that can be used in place of the standard maximum likelihood objective in training NMT models. This leads to a small improvement in training MT systems.

There were some concerns about the paper though: one was that the method itself seemed somewhat heuristic without a clear mathematical explanation. The second was that the baselines seemed relatively dated, although one reviewer noted that this seemed like a bit of a lesser concern. Finally, the improvements afforded were relatively small.

Given the high number of good papers submitted to ICLR this year, it seems that this one falls short of the acceptance threshold.